# Computational screening of the effects of mutations on protein-protein off-rates and dissociation mechanisms by τRAMD
Giulia D'Arrigo [1] ✉, Daria B. Kokh [1,4], Ariane Nunes-Alves [1,5] & Rebecca C. Wade [1,2,3] ✉

The dissociation rate, or its reciprocal, the residence time (τ), is a crucial parameter for understanding the duration and biological impact of biomolecular interactions. Accurate prediction of τ is essential for understanding protein-protein interactions (PPIs) and identifying potential drug targets or modulators for tackling diseases. Conventional molecular dynamics simulation techniques are inherently constrained by their limited timescales, making it challenging to estimate residence times, which typically range from minutes to hours. Building upon its successful application in protein-small molecule systems, τ-Random Acceleration Molecular Dynamics (τRAMD) is here investigated for estimating dissociation rates of protein-protein complexes. τRAMD enables the observation of unbinding events on the nanosecond timescale, facilitating rapid and efficient computation of relative residence times. We tested this methodology for three protein-protein complexes and their extensive mutant datasets, achieving good agreement between computed and experimental data. By combining τRAMD with MD-IFP (Interaction Fingerprint) analysis, dissociation mechanisms were characterized and their sensitivity to mutations investigated, enabling the identification of molecular hotspots for selective modulation of dissociation kinetics. In conclusion, our findings underscore the versatility of τRAMD as a simple and computationally efficient approach for computing relative protein-protein dissociation rates and investigating dissociation mechanisms, thereby aiding the design of PPI modulators.

Drug-target residence time has been observed to be a better indicator of the in vivo pharmacological activity of a drug than its binding affinity[1]. The residence time, τ, is the reciprocal of the dissociation rate constant, $k_{off}$, and describes the duration of the interaction, one of the key parameters determining biological function. Drug binding kinetics studies have been incorporated into many drug discovery campaigns, e.g., through quantitative structure kinetic relationship – (QSKR) models[2]. Moreover, off-rate screenings are becoming common for protein therapeutics, where they have shown promising results for the isolation of high-affinity antibodies[3,4] and it has been found that faster antibody off-rates can even lead to improved clinical safety and efficacy[5].

Protein-protein interactions (PPIs) are important pharmacological targets as their malfunction, which is often caused by mutations, is associated with numerous diseases, including cancer[6]. Many of these mutations

have been shown to affect the dissociation rather than the association rates, reinforcing the importance of the dissociation rate or off-rate as a key parameter determining molecular interactions and their pathophysiological roles[7–9]. Thus, there is a need for computational methods to predict the off-rates of protein-protein complexes. Moreover, besides predicting off-rates, these computational methods should enable the comprehension of the mechanisms governing dissociation processes in order to modulate them, engineer proteins with new functions, and design new therapeutics. Such dissociation processes occur on a wide-ranging timescale, from milliseconds to hours ($k_{off} < 10^{-3}\,s^{-1}$) for small-molecule drugs and extending to days ($k_{off} < 10^{-6}\,s^{-1}$) for protein-protein complexes. Such long times are beyond the reach of conventional molecular dynamics (cMD) simulation approaches, which are limited to the computation of residence times of maximally a few microseconds (or milliseconds for very small systems) with the most

[1]Molecular and Cellular Modeling Group, Heidelberg Institute for Theoretical Studies, Schloss-Wolfsbrunnenweg 35, 69118 Heidelberg, Germany. [2]Center for Molecular Biology (ZMBH), DKFZ-ZMBH Alliance, Heidelberg University, Im Neuenheimer Feld 282, 69120 Heidelberg, Germany. [3]Interdisciplinary Center for Scientific Computing (IWR), Heidelberg University, Im Neuenheimer Feld 205, 69120 Heidelberg, Germany. [4]Present address: CombinAble.AI, AION Labs, 4 Oppenheimer, Rehovot, 7670104, Israel. [5]Present address: Institute of Chemistry, Technische Universität Berlin, Straße des 17 Juni 135, 10623 Berlin, Germany, Berlin, Germany. ✉e-mail: giulia.darrigo@h-its.org; rebecca.wade@h-its.org

advanced resources[10]. Consequently, enhanced sampling methods are being developed to address this limitation and some of these have been used to study protein-protein dissociation, e.g., metadynamics, scaled MD, Gaussian accelerated MD (GaMD), weighted ensemble MD (WEMD), adaptive sampling to compute Markov state models (MSMs), and umbrella sampling (US)[10–14].

One of the most well-studied protein-protein complexes is that of the ribonuclease barnase (Bn) bound to its inhibitor barstar (Bs). It is considered a challenging target as it is one of the tightest known protein-protein complexes, with a residence time of ~75 h and $K_d$ on the order of $10^{-14}$ M in the wild type (WT) form[15]. The bimolecular association rate constants for a set of Bn-Bs mutants were computed in good agreement with experiment by Brownian dynamics simulations using a rigid body model of the proteins[16]. These simulations showed how complementary electrostatic interactions result in the very high on-rate of the WT proteins and also enabled the transient diffusional encounter complex to be structurally characterized[17] Although the structural differences between the apo and holo crystal structures of Bn and Bs are small, there are significant changes in a few interfacial sidechains that occur together with backbone adjustments that mean that a flexible protein model is necessary for computational docking of the proteins starting with the structures of the apo proteins[18,19] and for full characterization of binding and unbinding mechanisms. Using WEMD to calculate the rates for the binding process and the corresponding free energy landscape, Saglam and Chong found a similar binding mechanism to that reported for rigid-body simulations but more dynamic short-timescale motions than apparent in the crystal structures[18]. First attempts to reveal the unbinding of the Bn-Bs complex were made using steered molecular dynamics simulation (SMD) showing how the choice of the velocity and geometry of the force and its attachment point are of great importance for the final system conformation[20]. Later, US simulations were performed to compute the free energy profile of the dissociation process of the wild-type Bn-Bs complex and four mutants[21], and multiple-walker US simulations were employed to reveal multiple unbinding pathways[22]. Both the association and dissociation processes were observed by collecting 2 ms of simulation by using adaptive high-throughput cMD to compute a MSM[23]. Tempered binding simulations were also used to speed up the dissociation (reached in hundreds of microseconds for the Bn-Bs complex) and to capture the reassociation[24]. More recently, dissociation and rebinding events of Bn and Bs were captured by performing 2 µs of PPI-GaMD simulations that allowed free energy and kinetics calculations, in good agreement with experimental data and previous models[25]. However, many of the enhanced sampling methods require the use of collective variables, whose choice can differently affect the results and the interpretation of the mechanistic insights.

We recently developed τ-Random Acceleration Molecular Dynamics (τRAMD), a computationally efficient method to estimate the relative residence times of protein-small molecule complexes[26]. RAMD makes use of a randomly oriented force applied to the center of mass (COM) of the ligand to accelerate the dissociation of two molecular partners, speeding up the unbinding event to the nanosecond timescale[27]. The random choice of the force orientation means that no a priori assumption on the direction of ligand unbinding is made, allowing sampling of diverse unbinding pathways. Coupled with the our tool for the analysis of protein-ligand interaction fingerprints (MD-IFP), τRAMD allows the mapping of dissociation mechanisms, the detection of metastable states along unbinding pathways, and the identification of ligand-receptor contacts that influence the residence time[28]. τRAMD has been extensively validated and applied to a variety of small molecule-protein systems, ranging from soluble proteins[26,29,30] to membrane proteins such as GPCRs[31], showing good agreement with prior computational and experimental data, and demonstrating the broad applicability of the method. Compared to the dissociation of a small molecule from a protein, the computation of the dissociation rate of two macromolecules requires consideration of the generally flatter and larger intermolecular interaction interface and the greater number of conformational degrees of freedom of the dissociating macromolecules.

Here, we describe the adaptation and optimization of the τRAMD protocol and the MD-IFP tools for studying protein-protein dissociation. We then describe the application of the τRAMD methodology to a set of structurally and kinetically well-characterized protein-protein complexes: wild-type and 22 single and double-point mutants of barnase (Bn) and barstar (Bs)[15,33] (indicated with the suffix bn or bs when the mutation is in barnase or barstar, respectively), 6 single-point mutants of bovine β-trypsin (BT) and its bovine pancreatic trypsin inhibitor (BPTI), and wild-type and 6 single-point mutants of bovine α-chymotrypsin (BCT) and BPTI[34] (Fig. 1 and Supplementary Data 1). These systems display kinetic parameters that vary over orders of magnitude for both association rate constant and dissociation rate. The optimized τRAMD methodology provides good agreement between measured and computed residence times for this set of protein-protein complexes, suggesting that it can be used for estimating the relative residence times of diverse protein-protein complexes. Furthermore, it both recapitulates previously reported unbinding mechanisms and provides new mechanistic insights for the protein-protein complexes studied. It is noteworthy that previous studies of protein-protein dissociation mechanisms have focused primarily on extensive simulation studies of the WT proteins, with no or very limited analysis of mutants. Here, the τRAMD method makes multiple simulations of large numbers of protein-protein complexes computationally feasible, allowing for a comparative analysis of the dissociation kinetics and mechanisms of the investigated proteins and their mutants.

## Results
We first report the computed residence times and compare them with measured kinetic parameters. We then compare the dissociation pathways observed in the RAMD simulations with available experimental data and published simulations.

### Residence time estimation for the protein-protein complexes
**Barnase-Barstar.** The experimental $k_{off}$ values were taken from three different papers, two from Schreiber et al.[15,32] containing a large set of mutants, and one from Ikura et al.[33] containing a smaller subset (Fig. 1a and Supplementary Data 1). Notable differences between the two sets of measurements are observed, with the SPR-derived $k_{off}$ values from Ikura et al. being at least one order of magnitude lower (the WT $k_{off}$ value is even two orders of magnitude lower) than those obtained by stopped-flow measurements by Schreiber et al. Furthermore, between the publications from Schreiber from 1993 and 1995, the WT $k_{off}$ value differs by almost one order of magnitude. The residence times computed with τRAMD at a random force magnitude of 19 kcal/mol/Å with the five different methods tested are plotted against the experimental values in Fig. 2 and Supplementary Fig. S1. Overall, there is no notable variation between the computed values for the five different definitions of the dissociation event, with all showing a correlation with experimental values with an R2 of about 0.6. The slope is generally low at 0.14–0.19, although considering the time of the "by residue first" method yields a slightly steeper slope of the correlation curve and, simultaneously, larger error bars for some mutants. The slope becomes smaller when using a lower random force magnitude of 17 kcal/mol/Å, indicating lower sensitivity at lower random force magnitudes (Supplementary Fig. S2). However, the correlation with experimental data appears to be mutant-dependent with both the single and double mutants containing the K27Abn mutation showing relatively slower dissociation in the computations. Removing these mutants increases the correlation with experiment to an R2 of about 0.75 (Supplementary Fig. S3), suggesting that mutations that change the protonation pattern of the binding site are computed less accurately. The K27bn side chain points to E73bn, which has a deprotonated carboxylic acid group, but in all single and double mutants with the K27Abn mutation, E73bn was protonated as suggested by PDB2PQR. We therefore carried out a second set of simulations with the deprotonated form of E73bn for these K27Abn mutants. These showed no large difference for the K27Abn,

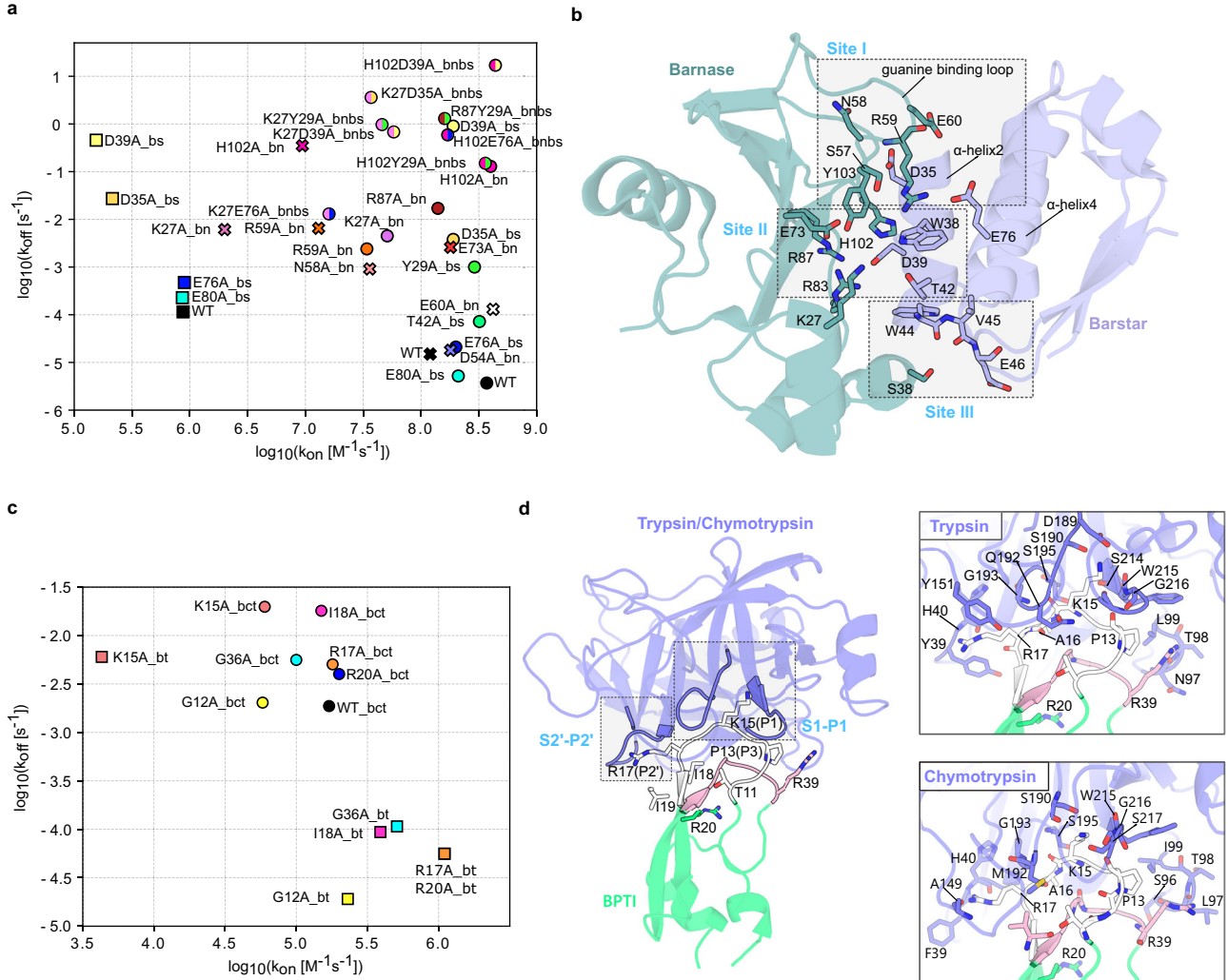

**Fig. 1 | Experimentally measured kinetic parameters and crystal structures for the complexes studied.** Available kinetic data and crystal structures are shown for the (**a, b**) barnase (Bn)-barstar (Bs) and (**c, d**) bovine trypsin (BT)-BPTI and bovine-chymotrypsin (BCT)-BPTI complexes. **a** The Bn-Bs kinetic data are from three papers indicated by symbols, where circle refers to ref. 15 and cross to ref. 32 (stopped-flow measurements), whereas the square is for ref. 33 (SPR measurements). Different colors indicate different single-point mutants whereas double mutants are indicated by two colors, with each half of the circle having the same color as the corresponding single mutant. **b** Bn-Bs crystal structure (PDB 1X1U[33]) with the three main interaction sites (Sites I, II, and III) and interacting residues labeled. **c** The

kinetic data for the BPTI complexes are from ref. 34 Squares and circles indicate BPTI mutants bound to BT and BCT, respectively. **d** Structures of BT (PDB 2PTC[41], shown for the full structure and the corresponding trypsin inset) and BCT (the equilibrated modelled structure, as described in the methods, is shown for the chymotrypsin inset) bound to BPTI. The S1-P1 and S2'-P2' interacting sites are highlighted. The primary binding loop (residues 11-19) and the secondary binding loop (34-39) of BPTI are colored white and pink, respectively. The two insets show the interacting residues in the BT-BPTI and BCT-BPTI structures. All measured kinetic parameter values and standard deviations are given in Supplementary Data 1.

K27E76Abnbs and K27D39Abnbs mutants but there was a larger difference for the K27D35Abnbs double mutant, whose residence time was overestimated when using the deprotonated form of E73bn, with the greatest difference for the "by residue first" method (Supplementary Fig. S4).

No further improvements in the correlation were found when comparing the RAMD-derived residence times to affinity (expressed as the inverse of $K_d$), yielding R2 values of 0.51 for both the COM-COM and "by residue first" methods. Remarkably, the contact-based bioinformatics tool for predicting binding affinity, PRODIGY[35], was not better at reproducing the experimental $K_d$ values, giving both lower correlation coefficients and sensitivity (Supplementary Fig. S5).

**BT and BCT -Trypsin inhibitor.** The experimental values for the two systems were all from Castro et al.[34] where $K_i$ and $k_{on}$ values were determined experimentally and $k_{off}$ values were derived using the equation $k_{off} = k_{on}/K_i$ (Fig. 1b and Supplementary Data 1). It is worth

noting that all the mutants of the BT and BCT systems (except for K15A) have very similar $k_{off}$ values, which lie in the same order of magnitude. The residence time was computed using the same two random force magnitudes (17 and 19 kcal/mol/Å) as for the Bn-Bs systems. For the BT-BPTI and BCT-BPTI systems, in contrast to Bn-Bs, the lower random force magnitude resulted in an overall better correlation with experimental values (Fig. 3 and Supplementary Fig. S6). At the random force magnitude of 17 kcal/mol/Å, all the BPTI mutants in the BCT-BPTI complexes were computed to have $\tau_{RAMD}$ residence times within the 5–20 ns range by all the criteria except for the "by residue first" method, which yields a better estimation for the slow K15A mutant (whose experimental $k_{off}$ is one order of magnitude lower) and better separates this mutant from the faster mutants, resulting in a R2 of 0.73 and a slope of 1.1 (Fig. 3). At the higher random force magnitude of 19 kcal/mol/Å, the mutants of the BCT-BPTI system have a wider range of computed $\tau_{RAMD}$ residence times values despite all having experimental $k_{off}$ values of $10^{-3} s^{-1}$, except for the K15A mutant ($k_{off} \sim 10^{-2} s^{-1}$), probably due to

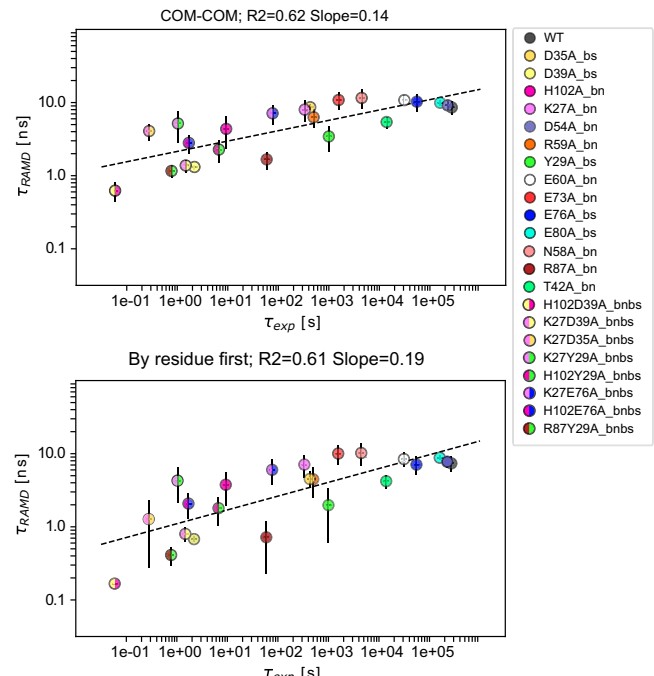

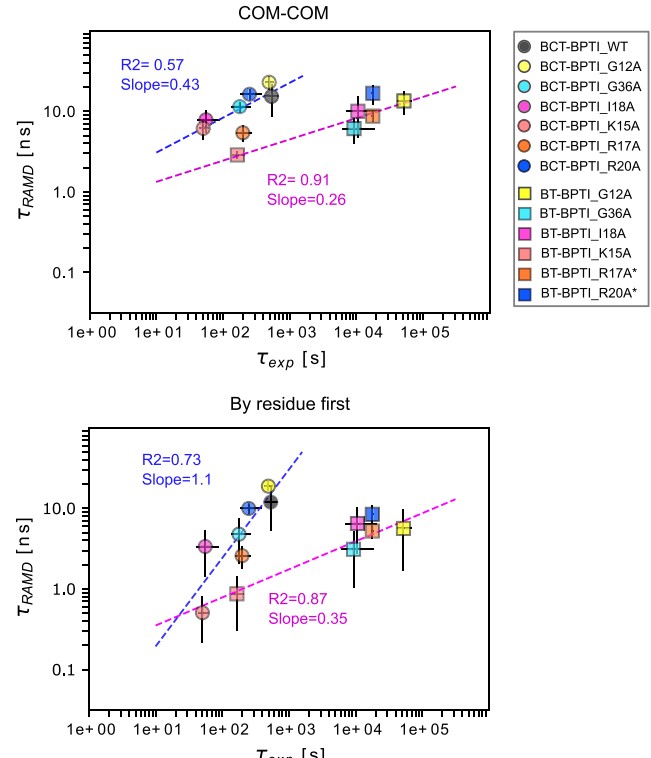

**Fig. 2 | RAMD residence time, $\tau_{RAMD}$, computed for wild-type and mutant Bn-Bs complexes vs experimental residence time ($\tau_{exp}$, inverse of $k_{off}$).** Results are shown for the ''COM-COM'' and ''By residue first'' definitions of residence time (see Methods), both giving a similar correlation with an R2 value of about 0.6. A random force magnitude of 19 kcal/mol/Å was used. The error bars indicate computed standard deviations and the dashed line shows the straight-line fit to all the data with the given R2 value and slope. Related data can be found in Supplementary Data 1 and on Zenodo[59].

**Fig. 3 | RAMD residence time, $\tau_{RAMD}$, computed for wild-type and mutant BT-BPTI and BCT-BPTI complexes vs experimental residence time ($\tau_{exp}$, inverse of $k_{off}$).** Results are shown for the ''COM-COM'' and ''By residue first'' definitions of residence time (see Methods). A random force magnitude of 17 kcal/mol/Å was used. The error bars indicate standard deviations and the dashed lines show straight-line fits, with the given R2 value and slope, in magenta and blue for the BT-BPTI and BCT-BPTI complexes, respectively. BT mutants indicated with * in the legend have experimental $k_{off} < 5.6 \times 10^{-5}$ s$^{-1}$ and were not used for determining the straight line fit. Related data can be found in Supplementary Data 1 and on Zenodo[59].

perturbations of the systems caused by the higher random force magnitude (Supplementary Fig. S7).

A good correlation and ranking between fast and slow mutants is achieved for the BT-BPTI systems at the random force magnitude of 17 kcal/mol/Å (R2 = 0.87–0.92, slope = 0.26–0.35). Conversely, at the higher random force magnitude of 19 kcal/mol/Å, the computed $\tau_{RAMD}$ residence times are all confined in a narrow window of around 1 ns, including the fastest dissociating K15A mutant (with a measured $k_{off}$ two orders of magnitude lower), leading to lower R$^2$ and slope values (Supplementary Fig. S7).

However, it should be noted that the straight-line fit for the BT-BPTI systems was only calculated for the four mutants G12A, G36A, I18A and K15A, as exact $K_i$ values could not be determined for the others because it was not possible to experimentally measure equilibrium dissociation constants below $5 \times 10^{-11}$ M[34]. Thus, no experimental error bars are provided for the other two mutants which are plotted but not included in computing the R2 values. The small dataset for the BT-BPTI complexes also lacks a $k_{off}$ value for the BT-BPTI WT complex as its value was determined in a different study from the mutants and was therefore not included in the correlation.

Similarly to Bn-Bs, both lower correlation and sensitivity were observed for the comparison of RAMD residence time to $K_i$ values (Supplementary Fig. S8). Prediction accuracy differed between the two systems, being lower for BCT-BPTI than BT-BPTI, but overall higher than the PRODIGY bioinformatics method which notably showed no correlation for the BT-BPTI mutants (R2 of 0.06).

**Comparison of the methods for computing RAMD residence time.** For both the Bn-Bs and BT-BPTI complexes, the correlation between the computed and the experimental values is similar regardless of the criterion chosen for computing the residence time and with a slightly greater

slope obtained when using the ''by residue first'' method. In contrast, for the BCT-BPTI complexes, an improvement in both the correlation and the slope is gained by using the ''by residue first'' approach due to the better estimation of the residence time for the fastest dissociating K15A mutant. Overall, the "by residue first" method seems to be more sensitive to very low residence times, and compared to the other tested approaches, generally tends to underestimate the residence time. According to its definition, the "by residue first" method extracts the residence time when the average distance between the interacting residues of the two proteins becomes higher than 5.5 Å, thus capturing the early unbinding events in the dissociation process. For some mutants, particularly the very fast dissociating ones, which are typically also highly unstable, this condition can be satisfied early in the unbinding simulation, thus resulting in lower computed $\tau_{RAMD}$ residence times. While not representing an issue for stable protein-protein complexes, like the ones we have tested, this could adversely affect flexible systems (e.g., protein-peptide complexes), leading to an underestimation of the residence time and in such cases, the other methods may be preferable.

## Determination of dissociation pathways for the barnase-barstar wild-type and mutant complexes

**Barnase-barstar mutations and interfacial interactions.** The complex between the extracellular ribonuclease, Bn, and its intracellular inhibitor, Bs, shows high charge complementarity at the binding interface, with the positively charged catalytic cleft of barnase (containing H102, R83, R87 and K27) facing the negatively charged α-helix2 of barstar (containing D35 and D39). All the mutations are located at the interface of Bn and Bs

(Fig. 1a). About 40 binding site residue contacts (having an average distance shorter than $d_{r-r} = 5.5$ Å) are observed in the crystal structures and in the equilibration trajectories, and all of these are conserved across the mutants except for those at the sites of the mutated residues (Supplementary Fig. S9A). Three different anchoring regions can be distinguished in the Bn-Bs complex (Fig. 1 and S9B). At site I, key interactions are hydrogen bonds between the Bn guanine binding loop (S57, N58, R59, E60) and D35 on the barstar α-helix2 and the salt bridge between R59 on Bn and E76 located on the Bs α-helix4. At the active site level (site II), interactions are mainly mediated by D39 on the Bs α-helix2 that makes hydrogen bonds with the Bn catalytic residues R83, R87, H102 and with Y103. At site III, K27 in Bn interacts with W38, D39 and T42 on the Bs α-helix2, and S38 in Bn makes hydrogen bonds with W44, V45 and E46 on Bs.

**Dissociation mechanisms.** For all of the analyzed mutants, the key interacting residues determining the predominant unbinding pathway are S57, N58, R59, E60 in Bn and N33, L34, D35 in Bs, with R59bn-D35bs being the longest lasting interaction. Starting from the bound complex with the three anchoring groups (clusters 1-3), the interactions evolve through the first loss of contacts mediated by S38bn and K27bs, together with the hydrogen-bonds between R83bn, R87bn and D39bs (clusters 5-6) (Fig. 4a, b). Following this, the hydrogen bonds of H102bn and Y103bn with D39bs are lost, and the contacts mediated by H102, Y103 and Q104 in Bn are overall reduced, leaving the group of contacts at the guanidine binding loop as the only anchoring site (cluster 7). Specifically, the carboxy oxygen of D35bs makes a hydrogen-bond to the backbone nitrogen of R59bn, E60bn makes hydrogen bonds with the sidechain of N33bs and the backbone of L34bs whereas R59bn additionally makes cation-π and ionic interactions with W38bs and E76bs, respectively. Among these contacts, the most populated is the hydrogen bond between D35bs and R59bn, which are in turn the residues that more strongly affect the kinetics of Bn-Bs.

A similar behavior is observed for most of the mutants, particularly for those having a long residence time, e.g., those with alanine mutations of the acidic residues D54 and E60 in Bn, and E76 and E80 in Bs (Figs. S10 and S11). On the one hand, D54bn and E80bs are not involved in the binding and their mutation leaves the overall pattern of interactions compared to the WT, as well as the overall stability of the complex, unchanged (Supplementary Fig. S12). On the other hand, the mutation of E76bs, despite depriving the complex of the salt bridge with R59bn, does not affect the establishment of the persistent contacts of R59bn with D35bs and W38bs. The small effect of the E76A mutation in Bs is also observed in the double mutants, K27AE76Abnbs and H102AE76Abnbs, where it hardly affects the residence time and the IFP content (Supplementary Fig. S10) compared to the single mutants K27Abn and H102Abn, indicating a minimal role in the dissociation kinetics of the Bn-Bs complex. Lastly, when E60bn is mutated, although the hydrogen bonds with L34bs and N33bs are lost, it is still able to bind D35bs via the backbone, thus maintaining one of the main interactions of D35bs. These observations indicate that the preservation of the key contacts mediated by D35bs and R59bn is crucial for prolonging the residence time.

Accordingly, the mutation of these two key residues to alanine has a comparable effect on the measured $k_{off}$ values, which increase by two orders of magnitude with respect to the WT, proving the key role of D35bs and R59bn in the dissociation kinetics of the complex. These mutations also notably affect the unbinding pathways. In the D35Abs mutant, the absence of the acidic residue disrupts the hydrogen-bonds with the Bn guanine binding loop, which is no longer pulled towards the Bs α-helix2. As a result, R59bn is not in a favorable position for the interactions with W38bs and E76bs, which are less likely to persist (Fig. 4e). Conversely to the WT, the hydrophobic contacts at site III (e.g., S38bn with W44bs and E46bs) are maintained almost up to the dissociation. Another pronounced difference is the lasting presence of the hydrogen-bonds mediated by R83bn and Y29bs (with Y29bs binding to the backbone of R83bn) which contribute to

reinforcing the interactions at the active site level, together with the H102bn-mediated contacts. The latter mainly drive the dissociation by determining the formation of an alternative dissociation route that goes through the residues from site II (i.e., H102bs-D39bn, H102bs-Y29bn, K27bs-D39bn) toward site III by means of S38bs-mediated contacts (Figs. 4e, f and S13).

Interestingly, mutation of D35bs severely affects the experimental residence time when in combination with K27Abn, reaching ~ms-s in the double mutant (roughly double the residence times of the corresponding single mutants D35Abs and K27Abn). K27 resides in the active site of Bn and makes one of the main interactions with D39bs in the bound state. Its mutation, likely overall weakening the interactions at site II, leads to an increase in the residence time of the same extent as D35Abs and R59Abn. The strong effect caused by the double mutant K27D35Abnbs further indicates the importance of the aspartic acid for the binding of the two proteins that, in such case, mainly take the alternative dissociation pathway through site III (Figs. S10 and S13).

Similarly, the mutation of R59bn results in dissociation via this alternative route. In this case, D35bs makes hydrogen-bonds with the backbones of R59Abn and E60bn and with the side chain of S57bn and is therefore still able to hold back the guanine binding loop. However, these contacts do not last long compared to the WT form, favoring the persistence of the Y29bn-mediated contacts which then gradually guide the dissociation through site III (Fig. 4h, I).

Among the single mutants, H102Abn and D39Abs affect the residence time most, leading to an increase in the $k_{off}$ of ~4 and 5 orders of magnitude, respectively, corresponding to the effect of most double mutants. As it is key in holding the two proteins together in the central core and is long preserved in the WT and many of the analyzed mutants, the absence of the H102bn-D39bs contact destabilizes the pattern of interactions at the active site level with a higher impact on the D39Abs mutant. In the latter, not only is the H102bn-D39bs hydrogen bond missing, but also the dependent contacts with R83bn and R87bn, altogether speeding up the detachment of the two proteins and the dissociation, which mainly follows the predominant path (Figs. S10 and S13). This behavior is further pronounced in the H102AD39Abnbs double mutant, which not surprisingly has the lowest residence time ($k_{off}$ higher by 6 orders of magnitude).

In summary, from these analyses, the hydrogen bond between R59bn and D35bs emerges as the key and most persisting interaction characterizing the WT and most of the mutants (Supplementary Fig. S14). In R59Abn, D35Abs and K27D35Abnbs, for example, the tight binding of the guanine binding loop to Bs is compromised and compensated by the contacts at the active site level and at site III that describe the alternative less populated dissociation route. However, mutations of the active site residues of Bn (e.g., H102A, D39A, K27A, R87A) more strongly affect the residence time, likely impacting the overall structural integrity of the active site (see higher RMSD in Supplementary Fig. S12), which simultaneously perturbs the stability of the hydrophobic contacts at site III. In this scenario, a restoration of the predominant path through site I is observed.

**Comparison of simulations.** Our simulations suggest the two loops surrounding the active site as the two main paths for unbinding. The same two dissociation pathways were identified by naïve multiple-walker US simulations of the WT Bn-Bs complex[22], with the route encompassing the guanine binding loop (i.e., residues 57–60 in Bn) being predominant, in good agreement with our findings.

Similar results were derived with adaptive high-throughput MD simulations and hidden Markov modelling of the WT Bn-Bs complex in ref. 23. In this study, the transition from the intermediate states to the bound state passes through a pre-bound state that, for 95% of the simulations has interactions of R59bn with E76, D35 and W38 in Bs and for the remaining 5% involves the hydrophobic contacts mediated by A37/S38bn and G43/W44bs. The bound state exists in the form of loosely and tightly bound states that are exchangeable. The loosely bound state, with 5% of the population, is highly flexible and stabilized by the interaction between K27bn and E80bs. The latter interconverts to the tightly bound state, with a higher population

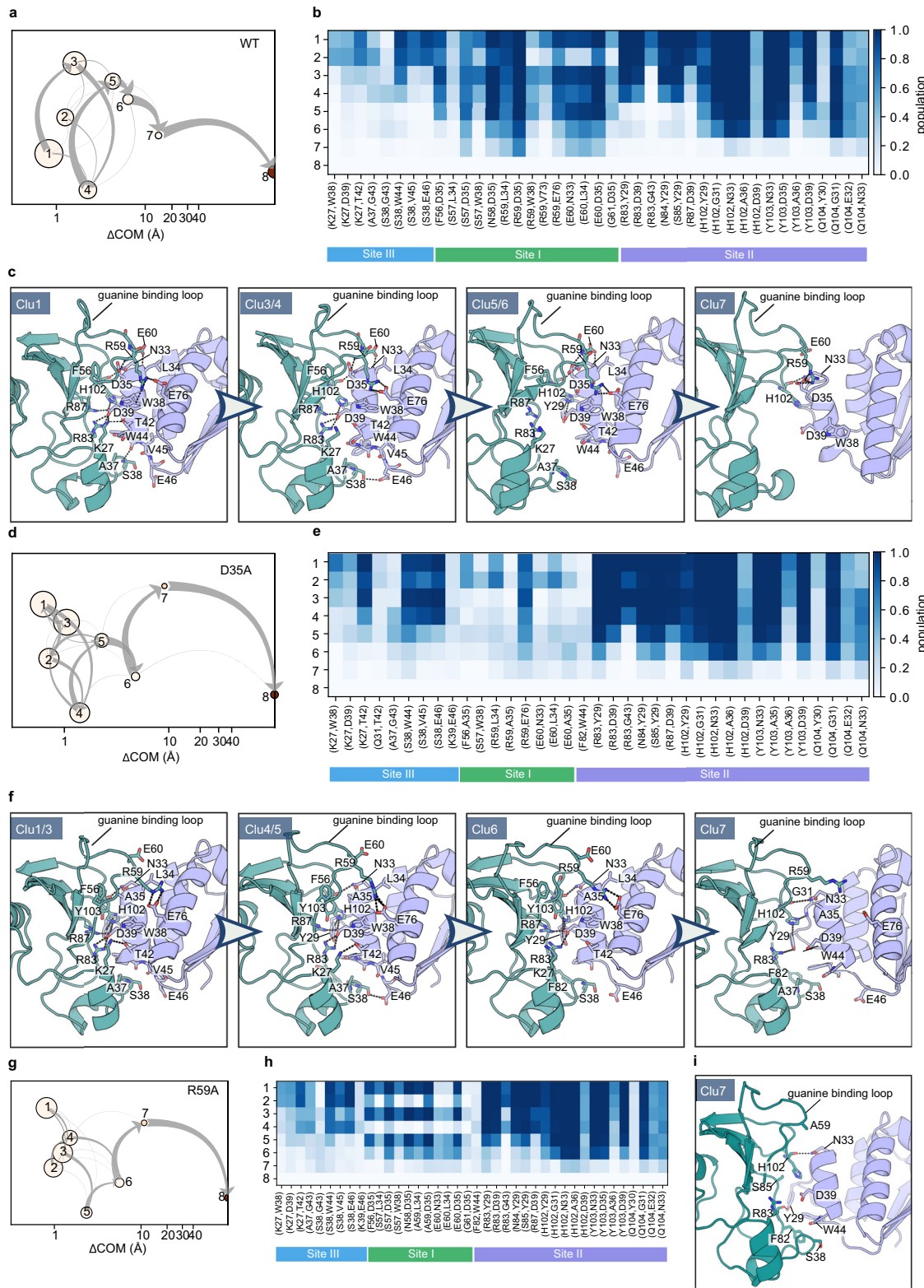

(95%) and stability, characterized by the salt bridge between R83bn and D39bs. These steps can be traced in reverse in our RAMD simulations. The contact between R83bn and D39bs is present in at least the first four clusters in the WT with occupancy of ~80% (Fig. 4a) and in most of the mutants (Supplementary Fig. S10). We do not observe the state where K27bn contacts E80bs because the distance is greater than 5.5 Å in our simulations but we observe a loosely bound state where K27bn contacts W38bs (cluster 1 in

Fig. 4a), which is in the vicinity of E80bs. Cluster 6 largely reproduces the pre-bound state with 95% population in ref. 23, with the interaction R59bn-D35bs having 70% occupancy of the frames before complete dissociation, while a very small occupancy is found for the S38bn-G43bs interaction. Moreover, hierarchical clustering of the pre-dissociation frames also shows that only a small number of trajectories have S38bn involved in the last contacts (Supplementary Fig. S15). WE simulations reported in ref. 36,

**Fig. 4 | Dissociation pathways in RAMD simulations of WT and mutant Bn-Bs complexes.** WT (**a–c**), D35Abs (**d–f**), and R59Abn (**g–i**). **a, d, g** Schematic representation of the clusters visited during the RAMD dissociation trajectories. Clusters are labeled and ordered by increasing mean COM-COM distance between proteins (x-axis). Cluster color indicates the averaged protein RMSD in the cluster from the starting structure. The gray arrows indicate the total flow between two clusters and their width increases with the number of trajectories having the corresponding transition. **b, e, h** IFP composition of the trajectory clusters resulting from the k-means clustering. Clusters are labeled from 1 to 8 (rows) and pairs of contacts are shown in the columns (the first residue index refers to barnase and the second refers to barstar). The population of each pair of residue contacts is shown with a color scale from blue (highest) to white (lowest). A legend bar indicating the sites (green for site I, violet for site II, and blue for site III) in the corresponding Bn-Bs complex interface is shown below the IFP maps. **c, f** Representative snapshots from the different clusters along the pathways from the bound to the unbound state with highlighting of the key residue contacts shown on a cartoon representation of the corresponding Bn-Bs complex. **i** Representative snapshot from cluster 7 indicating the important residue contacts during dissociation of the R59Abn complex shown on a cartoon representation of the corresponding Bn-Bs complex. Representative main and alternative dissociation trajectories for the WT and D35Abs complexes, respectively, are also shown in Supplementary Movie 1 and Supplementary Movie 2.

revealed a two-step binding process which, conversely to ref. 23, first goes through the contacts between S38bn and W44bs, and later dissociation of the latter and formation of the R59bn-D39bs contact in the encounter complex, which then favors binding between R59bn and D35bs and restoration of the hydrophobic contacts between S38bn and W44bs up to the complete bound state.

Recently, the clustering of six 2μs-long independent GAMD trajectories confirmed R59bn and A37-S38bn as the main sites for binding as well as the importance of long-range electrostatic interactions in the binding/unbinding of WT Bn-Bs[25]. GAMD simulations revealed a critical role in Bn binding and unbinding kinetics played by K27, R83, and R87, and more specifically by the charged residue pairs B27bn-E80bs, R59bn-D39bs, K27bn-D39bs, R83bn-D39bs, and R87bn-D39bs. This also aligns with our simulations where the active site residues (K27, R83, and R87 in Bn and D39 in Bs) emerge as crucial for the stability of the complex and are involved in key contacts until the formation of a loosely bound state.

**Role of water molecules at the interface.** Electrostatic interactions dominate the Bn-Bs complex resulting in a high number of polarizable water molecules at the interface that can be affected by mutation of the charged residues that are responsible for the electrostatic interactions. The role of waters at the interface of hydrophilic surfaces, like Bn-Bs, was previously described as a glue mediating interactions from the early stages of encounter complex formation and principally originating from interactions of charged residues[37]. This observation was supported by simulations of an artificial mutant in which the seven residues with the highest pairwise water-mediated connectivity were mutated to alanine (K27A, R83Q, R87A, H102A, in Bn, and D39A, Y29A, E76A, in Bs), showed a reduced water-mediated network compared to the WT complex [38]. In ref. 33 and ref. 39, crystallographic analysis of the wild-type Bn-Bs complex and mutants with single point mutations of four acidic residues, i.e., D39A and D35A in Bn, and E80A and E76A in Bs, revealed 31 water molecules within 3.4 Å of the two proteins and the presence of additional waters at the cavity created by the mutation.

In our study, we monitored the number of interfacial and buried waters and observed that water molecules are mobile during the equilibration simulations, periodically leaving and entering the interface, and that none of the studied mutations significantly affects the total number of interfacial water molecules, which was 30 ± 4 for the WT complex (Supplementary Fig. S15), even though the mutations resulted in rearrangements in the bridging and buried waters, causing the rupture of the stabilizing hydrogen-bonding network compared to the wild-type. Similarly, there was no clear difference in the number of buried water molecules during the replica equilibrations when comparing mutants that dissociate quickly or slowly. The WT complex has an average of 17 ± 2 buried water molecules and some mutants have more buried waters, e.g. R87Y29Abnbs with an average of 22 ± 2 water molecules, and others have fewer, e.g. D35Abs and K27D35Abnbs with an average of 14 ± 2 water molecules (Supplementary Fig. S16). No obvious pattern with respect to residence time was observed during RAMD dissociation trajectories for either interfacial or buried waters. However, it should be noted that the water model used for the simulations is not polarizable and may not fully account for the behavior of water in a highly polarizable environment such as the Bn-Bs interface.

## Determination of the unbinding pathways for the wild-type and mutant BT-BPTI and BCT-BPTI complexes

**BT and BCT inhibitor mutations.** BPTI forms very stable binary complexes with two serine proteases, BT and BCT. The two proteases are homologs with a sequence identity of 41.7% that have different substrate specificity and different affinities for BPTI. While BPTI coevolved with BT, thus developing a high affinity for it, BPTI binds to BCT because of its structural and sequence homology to BT, resulting in a weaker affinity complex[34]. At the active site, the negatively charged S1 pocket of BT preferably cleaves polar substrates (lysine or arginine), whereas the less polar BCT active site (with S189 in BCT instead of D189 in BT) preferably binds hydrophobic amino-acid residues[40].

Three major sites of interaction have been established in the two PPI complexes. The first is the primary binding loop of BPTI which encompasses residues 11 to 19 and is mainly responsible for the main chain-main chain contacts with the protease. The primary binding loop contains K15, also denoted as the P1 residue of BPTI, which is responsible for the majority of the contacts with the protease S1 pocket (residues 189–195, 214–220) (Fig. 1d). Part of this loop is a second site, the P2' site, represented by R17 which mainly interacts with the protease S2' binding cleft (residues 30–41, 151, and 192–193) through its side chain. The third main interaction spot is constituted by the secondary binding loop of BPTI, spanning residues 34 and 39, which contacts the protease residues 96–99.

Within the three interaction sites, there are differences between the two proteases. The key difference is at the P1 residue. In the crystal structure of the BT-BPTI complex[41], K15 in BPTI assumes a "down" conformation, extending to the bottom of the protease pocket and making a salt link with D189 and hydrogen bonds to the main chains of S190, S195 and G193 in BT (Figs. 1d and S9D). In the crystal structure of the BCT-BPTI complex[42], on the other hand, K15 assumes an "up" orientation with the side chain pointing away from S190 in BCT and towards the main chains of the protease and the inhibitor itself (Figs. 1d and S9F). At the P2' site of the BT-BPTI complex, R17 in BPTI is packed between Y39 and Y151 and binds the main chain of H40 in BT, whereas in the BCT-BPTI complex, the S2' cleft is wider due to the presence of a smaller residue, threonine, at position 151 in BCT and therefore, R17, being less packed, only makes interactions with the backbone of A149 in BCT. Finally, at the secondary loop, R39 in BPTI binds to N97 and L99 in BT in the BT-BPTI complex whereas, due to subtle modifications of the site in BCT, R39 interacts with the backbone of L97 in BCT in the BCT-BPTI complex.

In this study, we analyze the effect of 6 single point mutations in BPTI on the residence times of the complexes of BPTI with BT and BCT[34] (Fig. 1d). During the equilibration trajectories, the contacts between the proteins in the two complexes are well maintained for all the mutants, except the BPTI K15A mutant, which loses several contacts (Supplementary Fig. S9C, E).

### Dissociation mechanisms

**BT-BPTI mutants.** Among the three major sites of interaction, the primary binding loop on BPTI acts as an anchor to the protease pocket until shortly before the dissociation of the proteins, with the K15-mediated contacts, specifically to S190, G216, and G219, being the longest lasting ones, see Fig. 5.

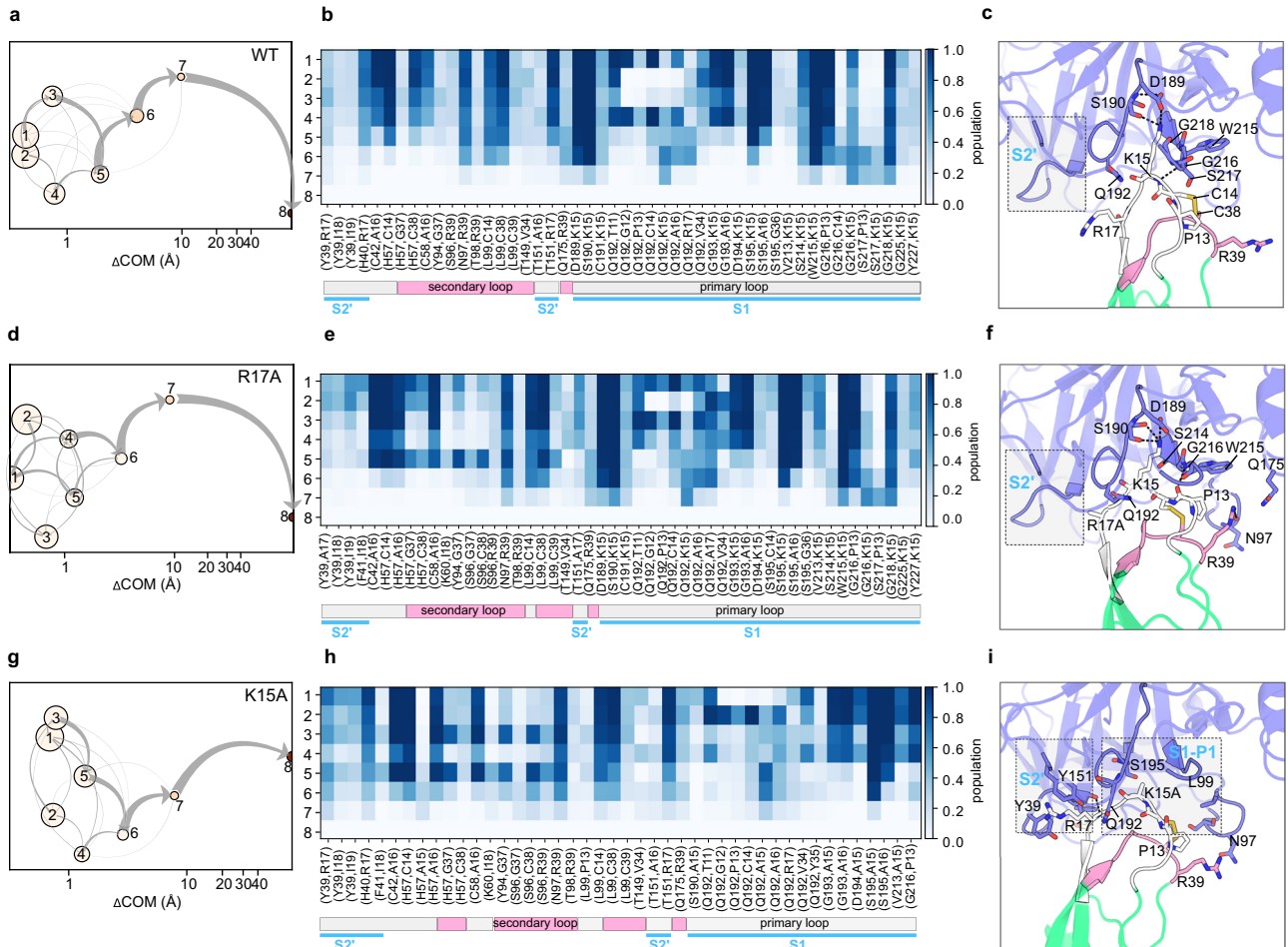

**Fig. 5 | Dissociation pathways in RAMD simulations of WT and mutant BT-BPTI complexes.** WT (**a**–**c**), BPTI R17A (**d**–**f**), and BPTI K15A (**g**–**i**). **a, d, g** Schematic representation of the clusters visited during the RAMD dissociation trajectories. Clusters are labeled and ordered by increasing mean COM-COM distance between proteins (x-axis). Cluster color indicates the averaged protein RMSD in the cluster from the starting structure. The gray arrows indicate the total flow between two clusters and their width increases with the number of trajectories having the corresponding transition. **b, e, h** IFP composition of the trajectory clusters resulting from the k-means clustering. Clusters are labeled from 1 to 8 (rows) and pairs of contacts are shown in the columns (the first residue index refers to BT and the second refers to BPTI. The population of each pair of residue contacts is shown with a color scale from blue (highest) to white (lowest). A legend bar indicating the sites and BPTI binding loops (primary: white; secondary: pink) in the corresponding BT-BPTI complex interface is shown below the IFP maps. **c, f, i** Important residue contacts during dissociation shown on a cartoon representation of the corresponding BT-BPTI complex. The primary binding loop (residues 11–19) and the secondary binding loop (34–39) of BPTI are colored white and pink, respectively. The S2' site is highlighted.

In the WT BT-BPTI complex, R17 in the P2' site of BPTI is close to Y39 of BT in the bound state but during dissociation alternates the cation-π interaction with Y151 and the hydrogen bond to the main chain of H40 in the protease. R39, in the secondary binding loop of BPTI, binds to S96, N97, T98, and L99 in the protease, with the R39-T98 hydrogen-bond being, together with the hydrophobic contact between the BT L99 and the BPTI intramolecular C14-C38 disulfide bridge, the most persistent until the dissociation. At the S1 site, interactions are dominated by the A16 main chain-mediated hydrogen bonds and by K15 of BPTI. K15 predominantly interacts with D189 (salt-bridge) and S190 (hydrogen-bond) at the bottom of the S1 pocket and with S214/W215 and G216/G218 at the entrance of the S1 pocket. Starting from the initial bound state with the three anchoring sites (clusters 1–5 in Fig. 5a, b), the dissociation evolves through the weakening of the R17 and A16-mediated contacts, prompting the detachment of BPTI at the P2' site: this can be traced by the increase of the hydrogen bonding between the R17 main-chain and the BT Q192 side-chain. Correspondingly, BT Q192 and G193 disengage, respectively, from T11/G12/P13/V34 and K15/A16 on BPTI, favoring newly formed interactions, such as S217-P13 and S217-K15, that are indicative of K15 gradually sliding from the S1 pocket to the P3 site (residues 12-14 of BPTI) (cluster 6). In the state just

prior to dissociation (cluster 7), the last interactions of BPTI K15 are made with Q192, S190, W215, G216 and G218 of BT (Fig. 5c).

The tendency to first detach from the S2' site is enhanced in the BPTI R17A mutant. Here, the hydrogen bond between the main chain of BPTI R17A and BT H40 is absent and, instead, contacts to Y39 and Y151 are present in the initial bound state. The interactions at the secondary binding loop site are more abundant than in the WT. The dissociation proceeds as for the WT by diminishing, up to complete disappearance, the interaction with Y39 and Y151 of the S2' cleft and gaining interactions mediated by the P3 site. As a result of the reinforcement of the contacts at the secondary binding loop, the loosely bound state that precedes the dissociation still has BPTI R39 interacting with the sidechain of Q175 and the backbone of N97 (cluster 7). As in the WT complex, the interactions between K15 and Q192, S190, W215, G216, and G218 are the very last ones and are most conserved during dissociation (Fig. 5d–f).

The importance of K15 in BPTI for the stability of the complex is demonstrated by the increase in $k_{off}$ of the K15A mutant compared to the WT complex of 5 orders of magnitude observed experimentally. When mutated, the pattern of protein-protein interactions drastically changes compared to the WT and the other mutants. The contacts established by the

K15 side chain (with D189, S190, S214, W215 and G216), which were the most conserved ones in the WT, are lost and the primary binding loop is now held solely by main chain-main chain interactions. Consequently, the contacts mediated by R17 (i.e., with Y39, H40, Y151) and the secondary binding loop are more persistent (Fig. 5g–i). Under these conditions, although K15A main chain-mediated contacts (i.e. with Q192 and S195) are preserved until unbinding in the majority of RAMD trajectories, thus representing the main route, a higher population of alternative routes going exclusively through R17 or R39 is present for the BPTI K15A mutant compared to the WT and other mutants (Supplementary Figs. S17-S18).

The numerous main chain-main chain interactions between the protease and the inhibitor require that the BPTI backbone conformation be preserved for its recognition at the BT site. Mutations of glycine residues on BPTI can cause a distortion of the backbone, impacting the overall stability and insertion into the protease pocket[34,43]. Substituting G36 in BPTI with an alanine introduces a large destabilization of the complex (as indicated by the higher RMSD than for WT during the equilibrations shown in Supplementary Fig. S19) in which the secondary loop of BPTI moved away from the corresponding site on BT and the β-sheet core of BT rotated compared to the WT (Supplementary Fig. S20A). Consequently, the interactions at the secondary binding loop, observed during RAMD simulations, are weakened in favor of reinforcement of those at the P2' site (Supplementary Fig. S21A). In addition, the steric hindrance caused by the presence of the methyl group of G36 A, which points to Q192 in BT, reduced the contacts of this residue with the primary binding loop (i.e., the contacts with T11, G12, and P13 are missing), overall perturbing the insertion of BPTI in the BT binding pocket. More importantly, the Q192-K15 contact, among the most conserved ones in the WT and other mutants, is absent in the G36A mutant bound state (clusters 1-5) and only appears in the loosely bound state, when K15 gradually leaves the S1 pocket (clusters 6-7), leading to dissociation (Supplementary Fig. S21A).

**BCT-BPTI mutants**. Despite the different protein-protein interaction patterns, no big differences in the dissociation pathways are observed for the BCT-BPTI complex compared to the BT-BPTI complex, and R17, R39 and K15 on BPTI remain the key interaction spots, with K15 responsible for the longest-lasting interactions along the dissociation trajectories (Supplementary Fig. S22).

The bound state (clusters 1-4 in Fig. 6a, b) differs from BT at all three main interaction sites. When packed within the S2' cleft, R17 of BPTI primarily makes a cation-π interaction with F39 and hydrogen bonds with the backbones of H40, A149 and M192, with the last interaction being the most conserved. In the secondary binding loop, R39 of BPTI makes a well-conserved hydrogen bond to the backbone of L97 and establishes a less persistent main chain-main chain contact with S96. While corresponding with BT in establishing lasting contacts with S190, G193, S195 and W215, K15 makes interactions with S217 and S218, representative of the "up" conformation, that are now firmly maintained. In addition, S218 forms well-preserved contacts with the primary binding loop. The first sign of detachment between the two proteins is a reduced packing of R17 within the S2' site, resulting in diminished contacts with F39 and H40, and simultaneously, the distancing of S96 from the secondary binding loop (cluster 5-6). In the final state preceding dissociation, R17 still contacts A149 and M192 while R39 binds to S97 right up until the complete separation of the complex (Fig. 6c).

Of the above-mentioned R17-mediated interactions, only the one with H40 and M192 remains at the S2' pocket in the R17A mutant and new hydrophobic contacts are made with L143. The primary and secondary binding loops interact more strongly with BCT and clearly drive the dissociation path (Fig. 6d–f).

Mutation of K15 causes a rearrangement of the loop forming the S2' pocket (containing A149) such that the loop is shifted away from BPTI and R17 exhibits a more pronounced insertion into the S2' pocket (Supplementary Fig. S20B). As a result, the protein-protein interaction pattern is modified, particularly at the P2' site where the nature and quantity of the

contacts are notably altered. While losing the hydrogen bonds to F39 and A149, R17 maintains interactions with H40 and makes new ones with Q73, W141, T151, P152 and G193 (Fig. 6h). K15A only establishes contacts with S195 and G193 which persist to the final state before dissociation (cluster 7, Fig. 6g–i). As for BT, dissociation can occur from either the P2' site or the secondary binding loop, with the former being highly populated as indicated by hierarchical clustering (Supplementary Fig. S17).

## Comparison of systems and simulations

In the RAMD simulations of BT and BCT in complex with BPTI, K15 in BPTI stands out as the residue responsible for the majority of the interactions and governing the dissociation of the proteins, although to a different extent in the two systems.

In BT-BPTI, the complete immersion of K15 in BPTI into the S1 pocket of BT generates an extended hydrogen bond network that promotes strong binding to the protease which, when disrupted by the K15A mutation, increases the dissociation rate by more than 2 orders of magnitude compared to the other BPTI mutants. On the other hand, the less polar S1 pocket in BCT and the less buried orientation of K15, with the correspondingly smaller network of hydrogen bonds, mean that this residue has less influence on the stability of the BCT-BPTI complex and there is a correspondingly smaller difference in $k_{off}$ between the K15A mutant and the rest of the mutants. Thus, K15 in BPTI is considered a hot-spot for the binding affinity to BT and a cold-spot for the binding affinity to BCT[44]. The observations on the different behavior of K15 in the two systems are also reflected in the number of water molecules at the interface (Supplementary Fig. S23). In the BT-BPTI complex, K15 establishes water-mediated bridges within the S1 site, and its mutation results in a slightly reduced number of interfacial waters (on average $11 \pm 2$ vs $14 \pm 2$ for the WT complex). Conversely, the less buried conformation of K15 in the BCT-BPTI complex and its reduced involvement in water bridges mean that the number of interfacial waters is hardly affected by the K15A mutation, (with on average $12 \pm 3$ water molecules vs $13 \pm 3$ for the WT complex). The complexity of the subtle interplay between the dynamic water network in the S1 pocket and the nature of residue 15 was also revealed by the results of crystallographic and simulation studies of the mutation of K15 in BPTI in the BT-BPTI complex to α-aminobutyric acid and increasingly fluorinated derivatives[45,46].

It is interesting to note that, across the BT-BPTI mutants, only the K15A mutation causes 2 orders of magnitude decrease in the measured association rate compared to the WT whereas the other mutants have comparable $k_{on}$ values[34]. The strong effect of the K15A mutation on both association and dissociation rates points to the essential role of K15 for the formation of the complexes as well as for their stability.

Kimura et al.[38] defined K15 as the key side-chain responsible for recognition between BT and BPTI and also investigated the role of R17, describing it as the latch that holds the high-affinity complex together. R17 also stands out for playing an anchoring role in our simulations where the disengagement of the arginine from the S2' pocket facilitates and initializes the dissociation process. This observation also agrees with a more recent study of the BT-BPTI system where US coupled to a Markov state model was used to recover the unbound, encounter and complex states and to describe the mechanism of binding[47]. In ref. 47, the difference between the unbound and bound states is mainly confined to the conformations of Y39 (BT) and R17 (BPTI), both part of the S2'-P2' interaction site: Y39 is preferentially oriented towards R17 which, in turn, should adopt a conformation to fill the S2' pocket and stabilize the bound state. In our RAMD simulations, the two above-mentioned residues are very close in the bound state and, as observed in IFP analysis, gradually move apart as the unbound state is approached. The Y39-R17 distance is therefore an indicator of the S2'-P2' site packing, whose loosening is the primer for unbinding.

## Discussion

In this study, we extend the application of τ-Random Acceleration Molecular Dynamics (τRAMD) from protein-small molecule[26,28–30] to protein-

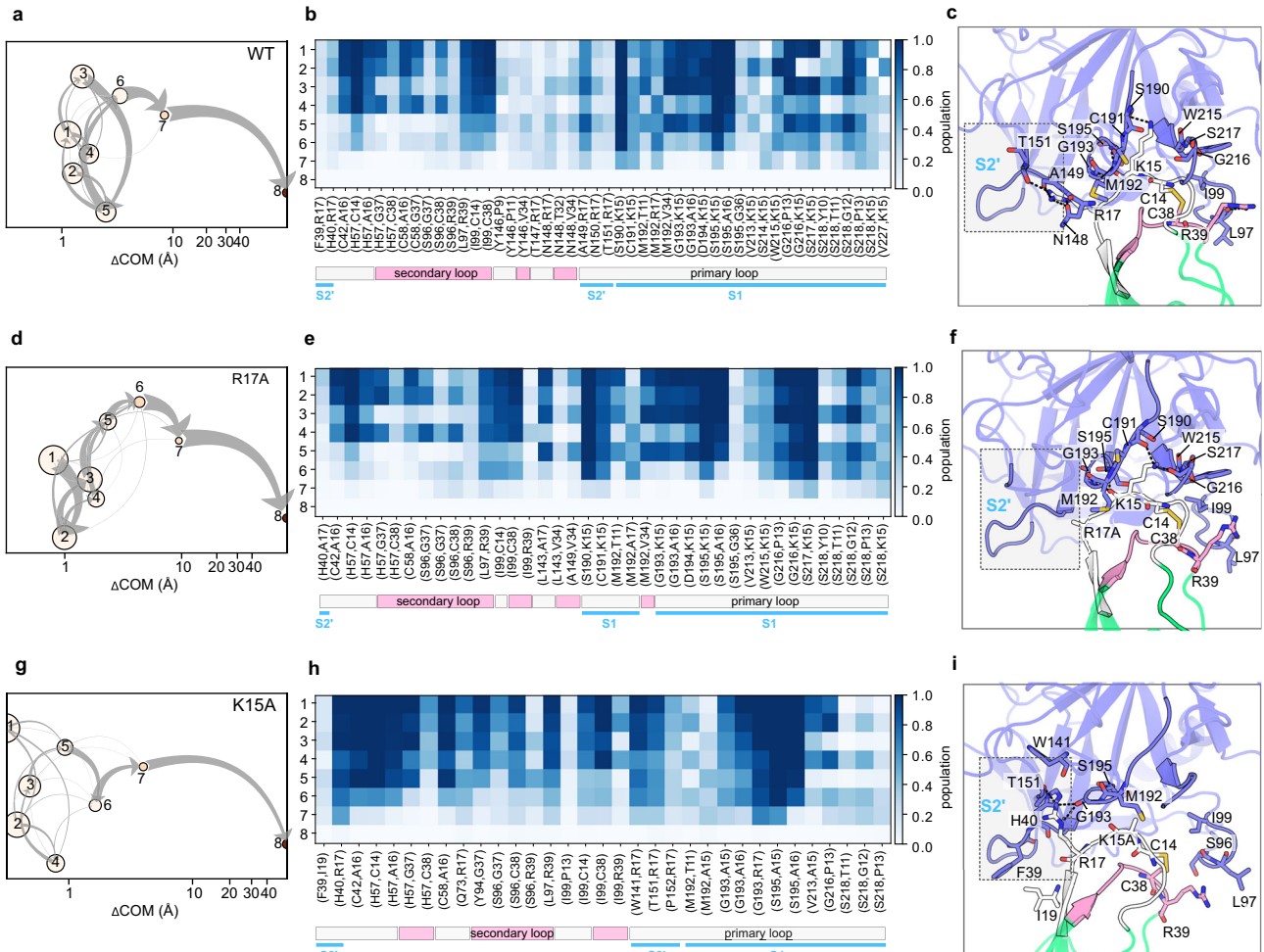

**Fig. 6 | Dissociation pathways in RAMD simulations of WT and mutant BCT-BPTI complexes.** WT (**a–c**), BPTI R17A (**d–f**), and BPTI K15A (**g–i**).
**a, d, g** Schematic representation of the clusters visited during the RAMD dissociation trajectories. Clusters are labeled and ordered by increasing mean COM-COM distance between proteins (x-axis). Cluster color indicates the averaged protein RMSD in the cluster from the starting structure. The gray arrows indicate the total flow between two clusters and their width increases with the number of trajectories having the corresponding transition. **b, e, h** IFP composition of the trajectory clusters resulting from the k-means clustering. Clusters are labeled from 1 to 8 (rows)

and pairs of contacts are shown in the columns (the first residue index refers to BCT and the second refers to BPTI. The population of each pair of residue contacts is shown with a color scale from blue (highest) to white (lowest). A legend bar indicating the sites and BPTI binding loops (primary: white; secondary: pink) in the corresponding BCT-BPTI complex interface is shown below the IFP maps.
**c, f, i** Important residue contacts during dissociation shown on a cartoon representation of the corresponding BCT-BPTI complex. The primary binding loop (residues 11–19) and the secondary binding loop (34-39) of BPTI are colored white and pink, respectively. The S2' site is highlighted.

protein complexes. In RAMD simulations, the unbinding event is observed within a few tens of nanoseconds, facilitating the fast and efficient computation of relative residence times. We tested the method on three protein-protein complexes and a wide range of mutants, totaling 36 different complexes.

Overall, τRAMD largely reproduces kinetic experimental data, showing system-dependent accuracy and sensitivity, which are generally lower compared to the protein-small molecule scenario. The lower accuracy can be attributed, in part, to the larger variation in dissociation times observed across different starting replicas, which often makes greater sampling necessary for protein-protein systems compared to protein-small molecule systems. Another contributing factor stems from the uncertainties inherent in the experimental data, mainly due to discrepancies in kinetic rates between different publications, different experimental techniques, as well as the small size of the datasets[48]. Furthermore, the accuracy of computed dissociation times may be influenced by the protonation state of the protein, which was kept constant during the simulations, but which is known to be able to change during the dissociation process, thus introducing an additional layer of

complexity and potential error into the simulations. However, the τRAMD procedure is able to reproduce the general trends and separate the fastest from the slowest mutants of a given system, and also reproduce the effect of mutations on PPI binding affinity, outperforming the state-of-the-art bioinformatics method[35].

The sensitivity of the method, surprisingly, did not improve by decreasing the RAMD force magnitude for Bn-Bs and it only marginally improved with the "by residue first" method. This method performs somewhat better than the other tested approaches in estimating τ, though with some tendency to underestimate the residence time. This suggests that the contributions to the RAMD residence time of the early unbinding events are more important than the later unbinding events for a correlation with experimental residence times of mutants.

When compared to other published methods, e.g., metadynamics, US, and PPI-GaMD, τRAMD is among the most efficient approaches. On the one hand, τRAMD can capture an unbinding event within a few tens of nanoseconds (in this study, dissociation of the slowest system was observed within 20 ns, meaning a maximum of ~1.5 μs of total simulation time was required for ~75 RAMD trajectories of each mutant)

compared to e.g., PPI-GaMD which required six independent 2 μs simulations to generate a total of 19 dissociation and 16 rebinding events for WT Bn-Bs[25]. On the other hand, a big advantage is that no pathway or collective variables for unbinding need to be defined before running RAMD simulations, and therefore, the method is not biased towards any particular dissociation pathway and is straightforward to use.

The unbiased nature of RAMD facilitated the capture of a variety of unbinding routes that could be linked with the residence times of the mutants, further helping in the differentiation of fast and slow mutants of the same system from a mechanistic viewpoint. The interaction fingerprint analyses of RAMD simulations were key for unraveling the mechanisms of dissociation and, importantly, pinpointing the hotspots that could be targeted to selectively modulate protein-protein interactions. In the case of Bn-Bs, site I emerges as responsible for the most prolonged interactions, steering the primary unbinding route (Fig. 7a). Conversely, dissociation from site III is linked to fast-dissociating mutants. In the BT-BPTI and BCT-BPTI complexes, residues K15, R17, and R39 are identified as the three primary anchoring spots for sustained interactions, characterizing the dissociation pathways (Fig. 7b). The mechanistic details of the unbinding process are in general agreement with previous studies of the WT complexes, further supporting the reliability of the τRAMD method and highlighting its potential to investigate protein-protein dissociation.

In conclusion, τ-RAMD is a simple and efficient computational screening method applicable to studying the effects of PPI mutations on off-rates. Its application can be extended from small molecules to biologics for the purpose of estimating dissociation rates and investigating dissociation mechanisms. We anticipate applications in the assessment of the stability of predicted protein-protein interfaces, in protein design and mutant screening, and in the elucidation of how protein-protein binding kinetics influence biochemical and signaling networks in healthy and diseased states.

## Methods

### System preparation

**Barnase-Barstar**. 3D structures of the WT and 22 mutant Bn-Bs complexes were generated using the most complete available crystal structure (PDB ID: 1X1U[33], resolution 2.3 Å, chains C (Bn) and F (Bs)) and mutations to alanine were performed by just deleting atoms. The Bs double cysteine mutation (C40A/C80A) present in the crystal structure was retained as it was used for all the experiments[15,32,33]. All crystallographic water molecules from chains C and F were retained. Systems were protonated at pH 7.5 (corresponding to the experimental conditions – see below) using PDB2PQR[49].

**Beta-Trypsin and Alpha-Chymotrypsin -Trypsin inhibitor**. The structure of the WT BT-BPTI complex (PDB ID: 2PTC[41], resolution 1.9 Å) was used to generate the structures of the WT and the 6 mutant complexes. For BCT in complex with the inhibitor (BCT-BPTI), the structure from PDB ID 1T8O[42] (resolution 1.7 Å) was used to generate the structures of the WT and the 6 mutant complexes. The missing loops in this structure were modelled using Chimera[50] and Modeller[51] using default parameters. As K15 in BPTI is mutated in this structure, the WT BPTI structure was taken from PDB ID 2PTC[41] and superimposed. All crystallographic water molecules were retained. Protons were assigned at pH 8, the pH of the experimental measurements, using PDB2PQR[49].

### Setup and equilibration

All the systems were set up according to the following protocol. The AMBER ff14SB force field[52] was used and the systems were solvated with TIP3P[53] water molecules using a periodic cubic box with a margin of 25 Å to ensure correct treatment of periodic boundaries up to the dissociation event. Na$^+$ and Cl$^-$ ions were added to neutralize the systems and immerse them in a salt solution of 75 mM and 22 mM ionic strength for the Bn-Bs and BT/BCT-BPTI systems, respectively. Energy minimization, heating and equilibration

**Fig. 7 | Representative protein-protein dissociation pathways.** Comparison of the bound state and the predominant and alternative dissociation routes for the WT Bn-Bs complex and its mutants (**a**) and for the BT/BCT-BPTI complexes and their mutants (**b**). **a** Key interacting residues along the unbinding paths through sites I and III are labeled and shown as spheres (in grey for Bn and salmon for Bs). These are also observable in Supplementary Movie 1 and Supplementary Movie 2. **b** The key residues on BPTI along the unbinding pathways are shown as white spheres and labeled.

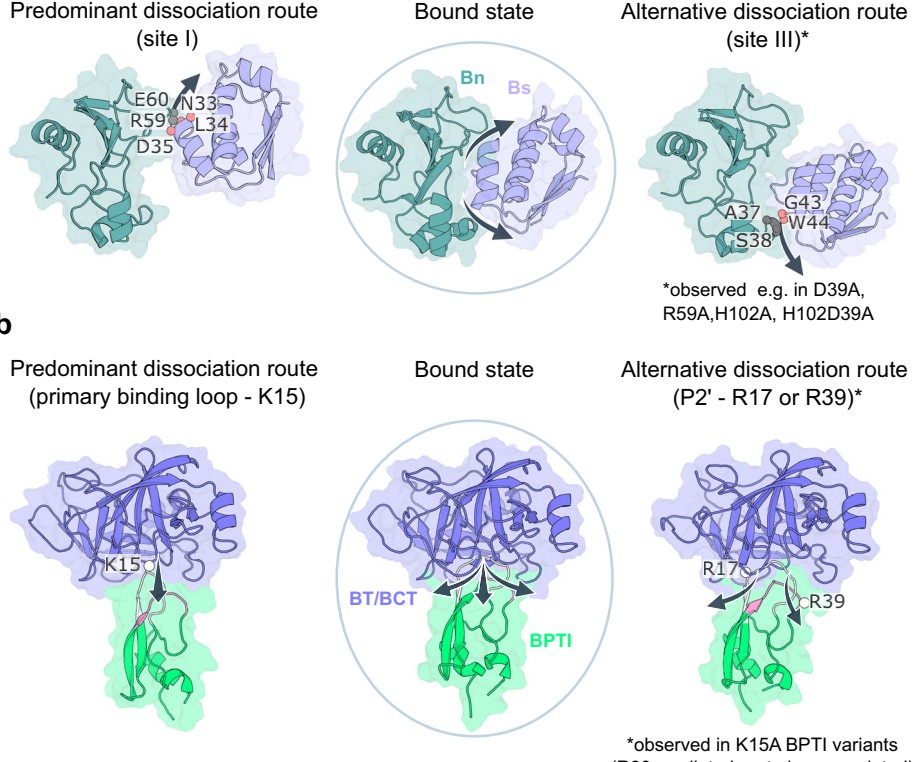

were performed using the AMBER14 software[54]. Systems were initially energy minimized in four stages, each consisting of 1000 steps of steepest descent and 500 steps of conjugate gradient minimization, using restraints on all heavy atoms that were decreased at each stage from 500 to 100 to 5 and then to 0 kcal mol$^{-1}$ Å$^{-2}$. Then, with restraints of 50 kcal mol$^{-1}$ Å$^{-2}$ on the heavy atoms, heating to 300 K was carried out stepwise in the NVT ensemble (Langevin) for 200 ps followed by equilibration at 300 K and 1 atm in the NPT ensemble (Langevin thermostat and Berendsen barostat) for 1 ns. Then a second equilibration step was carried out in the NPT ensemble without any restraints for 1 ns. For all simulations, a cutoff of 10 Å was used for nonbonded Coulombic and Lennard-Jones interactions, and periodic boundary conditions with a Particle Mesh Ewald treatment of long-range Coulombic interactions were used. A 2 fs time step was employed with bonds to H atoms constrained using the SHAKE algorithm[55]. The equilibrated systems were then used to generate the input starting structures for the τRAMD protocol implemented in GROMACS[56] (GROMACS-RAMD code available at https://kbbox.h-its.org/toolbox, version 1.0 was used). First, the coordinates and topology files were converted from AMBER to GROMACS format using ParmEd[57], and then a short simulation of 10 ns was run in the NVT ensemble (with a Berendsen thermostat). From this, the last snapshot was used to generate at least 5 trajectories of 20 ns duration in the NPT ensemble (using a Nose-Hoover thermostat and a Parrinello-Rahman barostat) with velocities either taken from the previous step or generated from the Maxwell distribution to ensure variability. The final coordinates and velocities from each trajectory were then used to initiate RAMD simulations under the same NPT conditions.

## RAMD simulations

The τRAMD protocol, previously described in refs. [28,31], was adjusted for application to protein-protein complexes. A random force was applied to one of the two protein partners, typically the smallest protein, although test simulations showed that the dissociation time was similar if the force was applied to the larger protein. The random force was applied to the COM of the protein and distributed to all its atoms via mass-weighting. The displacement of the protein was checked every 100 fs and, if it was below 0.025 Å, a new orientation of the force was generated randomly. Simulations were stopped when the protein-protein COM separation reached 70 Å. Each of the five starting configurations from the final snapshots of the equilibration simulations was used to generate at least 15 RAMD dissociation trajectories, each of up to 40 ns duration. Snapshots were recorded at intervals of 10 ps. The considerably lower protein-protein dissociation rates required the use of a higher random force magnitude than that typically employed for protein-small molecule complexes (14 kcal/mol Å). Here, force magnitudes of 17 and 19 kcal/mol Å were used for Bn-Bs and the two BPTI-bound systems, respectively. The force magnitude was assigned considering the slowest dissociating mutant for each system.

## Trajectory analysis

### Protein-protein contact analysis.

Protein-protein residue pairs (PP-REs) were defined as all combinations with one residue from one protein and one from the other within a distance of 15 Å. PP-REs were computed for each frame for both the equilibration replicas and the RAMD simulations as follows:

1. Protein atoms were grouped based on potential interactions, including aromatic, hydrophobic, hydrogen bond donor/acceptor, and cationic/anionic interactions (see Supplementary Table S1).
2. For each PP-RE, the COM-COM distance between the groups of atoms defined in step 1) and the COM-COM distance of the side chain carbons, or the backbone atoms in the case of Gly, were computed.
3. The minimum of the distances computed in step 2) was taken as representative of the PP-RE.

For every PP-RE, the final distance values were stored, for each snapshot, in a matrix that also contained additional information such as the protein root mean squared deviation (RMSD) and center of mass

(COM) position. Binding site contacts (BS-REs), i.e., close contacts determining the interaction, were extracted from the PP-REs matrix and were defined as the pairs of contacts whose distance was less than a threshold distance $d_{r-r} = 5.5$ Å for over half the length of the equilibration trajectories.

### Residence time estimation.

Different criteria for computing RAMD residence times of the protein-protein complexes were explored. For the standard τRAMD protocol, the unbinding time was recorded when the COM-COM distance exceeded 70 Å and the RAMD trajectory was stopped or when the maximum trajectory time (40 ns) was reached. For the other criteria, PP-REs occurring within the threshold distance $d_{r-r}$ for binding site contacts were considered and the dissociation time was defined by one of the following four criteria: (A) the first frame in which the average distance between the BS-REs was longer than $d_{r-r}$ ('by residue first'), or (B) the last frame in which the average distance between the BS-REs was shorter than $d_{r-r}$ ('by residue last'), or (C) the last frame in which the number of contacts was greater than 50% of BS-REs ('many contacts last'), or (D) the first frame in which the number of contacts was less than 50% of BS-REs ('few contacts first'). For each criterion, after collecting the dissociation times from each set of RAMD trajectories that originated from the same replica, the time for half of the trajectories to dissociate, corresponding to 50% of the cumulative distribution function, was computed (Supplementary Fig. S24). A bootstrapping procedure (50000 rounds with 80% of samples selected randomly) was then applied to obtain the residence time for each replica, $\tau_{repl}$, with the corresponding standard deviation, $SD_{repl}$. These values were then averaged over all the replicas to give the RAMD residence time, $\tau_{RAMD}$, and $SD_{RAMD}$. If $SD_{repl}$ or $SD_{RAMD}$ exceeded 50% of the corresponding residence time value, the number of simulations per replica or the number of replicas, respectively, was increased.

### Clustering.

To detect the most visited regions before dissociation and the pathways of dissociation, we conducted an MD-IFP clustering analysis in the protein interaction fingerprint space. The clustering analysis was performed on the part of RAMD trajectories satisfying the dissociation criterion based on the "by residue first" method, namely frames ranging from the initial loosely bound state until trajectory termination, plus 200 frames (2 ns) preceding this time window to ensure coverage of the bound state. If a trajectory did not satisfy the "by residue first" criterion, i.e., it did not dissociate, it was not considered for the analysis. From the selected RAMD frames for all the replicas, the corresponding PP-REs matrices were merged and converted into a single binary contact matrix where contacts shorter than $d_{r-r}$ were set equal to 1 and those longer were set to 0. The derived contact matrix was then converted into a distance matrix using the Jaccard distance $d_J$, a coefficient that quantifies the dissimilarity between two sets. For each pair of RAMD frames, the associated $d_J$ was given by the ratio of the number of common PP_RE contacts to the total number, subtracted from 1. As a result, a $d_J$ of 0 was associated with RAMD frames having the maximum identity in PP_REs content and vice versa for a $d_J$ of 1. The derived distance matrix was finally clustered using the *k-means* algorithm implemented in the scikit-learn package[58] with default values for parameters. The number of clusters to generate was derived from a balance between assessment with the Silhouette method and the most suitable number considering all the mutants of the same system. Clusters were then plotted and positioned on an increasing logarithmic scale of the average protein COM displacement (ΔCOM) in the cluster from the starting frame, reflecting dissociation from the bound (small ΔCOM) to the unbound state (large ΔCOM). Each cluster contains frames with similar PP_REs content, and the size of the cluster indicates its population. Considering two clusters (Ci and Cj), the transition from one to the other, i.e. (Ci → Cj) - (Cj ← Ci), is indicated by arrows which collectively indicate the routes to unbinding. For every cluster, the corresponding PP-REs contacts were retrieved from the binary matrix and shown as a heatmap indicating the degree of

occurrence in the clusters on the transition from the bound to the unbound state.

Hierarchical clustering of the pre-dissociation frames with fewer than three contacts was then performed to identify the last interactions during the dissociation process and more easily extract and characterize the different unbinding pathways. Protein structures were visualized using the PyMol Molecular Graphics System, version 2.4.1 (Schrödinger, LLC).

**Monitoring of water molecules**. Water molecules at the interface were monitored during the equilibration trajectories of the Bn-Bs and BT/BCT-BPTI complexes. All the water molecules at the interface were detected (defined as having their oxygen atom within 3.5 Å of any heavy atom of Bn and Bs), including the buried waters (for Bn-Bs defined, on the basis of the crystal structure, as having their oxygen atom within 3.5 Å of an oxygen atom of residues D35bs and D39bs, the OD2 atom of D54bs, or the backbone N or O of residues L42bn, R83bn, D35bs and V45bs).

**Processing of experimental data**
Experimental measurements of $k_{off}$ values for the Bn-Bs systems were retrieved from three different publications, each reporting different techniques and experimental conditions used[15,32,33]. In ref. 32 and ref. 15, Schreiber et al. performed stopped-flow measurements using 100 mM NaCl at pH 8, and 50 mM Tris-HCl at pH 8, respectively, while in ref. 33 Ikura et al. performed surface plasmon resonance (SPR) experiments using 150 mM NaCl at pH 7.4. Therefore, for each series, the $k_{off}$ values were normalized to the $k_{off}$ of the WT complex and then multiplied by the $k_{off}$ for the WT Bn-Bs complex from ref. 15. The final $k_{off}$ value was obtained by averaging the normalized values for each mutant (Supplementary Data 1 and Fig. S2). The same procedure was applied to $k_{on}$ values. From the averaged $k_{off} \pm SD$ and $k_{on} \pm SD$, $K_d$ was calculated through the relationship $K_d = k_{off}/k_{on}$ and the SD derived by error propagation (Supplementary Data 1). The experimental values for the BT- and BCT-BPTI systems were derived from one paper[34] where $k_{on}$ and $K_i$ values (mean value $\pm$ SD) were determined experimentally by spectrophotometric assays with 1 mM HCl and 20 mM $CaCl_2$ at pH 8.2. The corresponding dissociation rates were then calculated through the relationship $k_{off} = k_{on}/K_i$ and the SD of the derived $k_{off}$ values was obtained by error propagation (Supplementary Data 1).

**Statistics and reproducibility**
The experimental data used in this study were processed as described in the previous section. The molecular dynamics simulations were performed in replicas as described above with the number of replicas being dependent on the values for the standard deviations of the residence times obtained by bootstrapping. The analysis of the trajectories by clustering was assessed Silhouette method as described above.

**Reporting summary**
Further information on research design is available in the Nature Portfolio Reporting Summary linked to this article.

**Data availability**
Input data (PDB and parameter files) and scripts for running simulations, representative RAMD trajectories and data for computing residence times, computed interaction fingerprints, and Jupyter Notebooks for reproducing the analysis are available at Zenodo[59]. Source data for experimental and computed residence times can be found in Supplementary Data 1. All other data are available from the corresponding author on reasonable request.

**Code availability**
All software used for the simulations is freely available (PDB2PQR (https://server.poissonboltzmann.org/pdb2pqr), ParmEd (https://github.com/ParmEd/ParmEd), GROMACS version 2020.3 (www.gromacs.org), GROMACS-RAMD version 2020.3-1.0 (https://github.com/HITS-MCM)) or available with an academic or commercial license (AMBER14 (https://

ambermd.org/)). Code scripts (tauRAMD and MD-IFP) and Jupyter notebooks used for data processing, analysis, and plotting are available at Zenodo[59] and the MD-IFP scripts (Protein-protein version 1.1) are also available at https://github.com/HITS-MCM. Codes are written in Python v. 3.x and tested on Python v. 3.8.5. PyMol v. 2.4.1 (https://www.pymol.org/) was used for molecular visualization and figure-making and is available with an academic or commercial license.

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

## Acknowledgements
We thank the European Union's Horizon 2020 Framework Programme for Research and Innovation under Grant Agreement 945539 (Human Brain Project SGA3), and the Klaus Tschira Foundation for financial support. We thank Bernd Doser for the RAMD implementation in Gromacs and Stefan Richter for technical support.

## Author contributions
G.D.A.: conceptualization, methodology, software, validation, formal analysis, investigation, data curation, writing—original draft, writing—review and editing, visualization. D.K.B.: conceptualization, methodology, software, formal analysis, investigation; A.N.A.: conceptualization; R.C.W.: conceptualization, writing—review and editing, supervision, funding acquisition.

## Competing interests
The authors declare no competing interests.
