## [Peer Review File · Communications Biology]

Reviewers' comments:

Reviewer #1 (Remarks to the Author):

This manuscript applied τ RAMD to investigate protein-protein unbinding pathways and estimate dissociation rates of the complexes. The work also compared τ RAMD, GaMD and metadynamics, and showed that τ RAMD is more computation efficiency. The authors performed pretty thorough tests with three protein-protein complexes: ribonuclease barnase (Bn) and barstar (Bs) wild-type and 22 single and double-point, 6 single-point mutants of bovine β -trypsin (BT) and its bovine pancreatic trypsin inhibitor (BPTI), and wild-type and 6 single-point mutants of bovine-chymotrypsin (BCT) and BPTI. The authors also collected 3 different exp from Bn-Bs and normalized them for easier comparison with computation results. The computation method is sound, and results are informative.

The tests for these extensive mutant datasets are useful for understanding the mutation effects and also demonstrated that τ RAMD's performance is impressive. The agreement between 28 computed and experimental data are very good. The method is robust and efficient. The reviewer found that the order of the manuscript is a bit hard to follow. It would be nice to see that τ RAMD yielded the same trend as experimentally measured koff earlier. The results nicely validate the method, but the reviewer had to find the figure near the end of the manuscript. Since the authors discussed the dissociation mechanisms, it would be important to show first that the computation agrees with experimental results. And then the discussion of the mechanisms would be more meaningful. It is fine to discuss the residence time later. but it may be helpful to move Figures 5 and 6 (either reporting koff or 1/koff) to near the beginning of Results, so it can be seen easily.

It is interesting to read that there are anchoring groups, e.g. clusters 1-3 in Figure 2A. However, what are these clusters exactly? Figures like 2A provides some information. But without more figures, it's hard to learn how they dissociate exactly and why these are important. Some representative figures for the pathways and/or the key clusters can be very useful.

Minor points.

line 75, what is WEMD?

line 98 abbreviation IFP appeared before the full name.

recommendation: publish after some revision.

Reviewer #2 (Remarks to the Author):

In their article „Computational screening of the effects of mutations on protein-protein off-rates and dissociation mechanisms by τ RAMD”, D’Arrigo et al. present the application of tauRAMD for the calculation of relative protein-protein unbinding rates. The authors test the approach on three different protein complexes and various mutants thereof.

While tauRAMD has initially been designed for the ranking of protein-ligand unbinding rates, the application for much more involving, and definitively represents a method development that deserves publication. As the authors state, such a tool is of interest for e.g. designing protein therapeutics with defined binding and unbinding kinetics. Performing a benchmarking with three different complexes and their mutants is commendable.

In general, I think the research presented is sound, and the manuscript is definitively worth publishing in CommsBio. However, I find some parts hard to read, especially the description of unbinding pathways, which appears to me quite lengthy and highly detailed, and will most likely only be of interest for experts working exactly on the three presented protein/protein complexes. Given that tauRAMD is a method to firstly calculate unbinding times, I would like to ask the authors to restructure the manuscript and to first present the results on unbinding times. After having obtained the information of how well tauRAMD can predict such times, the interested reader then can delve into the intricacies of the unbinding paths (which might be needed to be shortened) and the role of water molecules therein.

Specific remarks

p. 5, line 97: the formulation “no bias in the directions of the unbinding pathways” is misleading, as tauRAMD applies a bias force. I assume the authors want to state that no assumptions on the unbinding path need to be made beforehand.

p. 8: Please point out that the pH values for protonation states were being chosen at 7.5 and 8 for reproducing conditions in the reference experiments.

p. 10, line 191: how was the random force applied to the proteins? As a force on the center of mass that was then distributed to all atoms via mass weighing?

p. 12, line 249: please explain shortly what the Jaccard index is.

p. 22, 2nd paragraph: I think the authors need to be careful with their results on water motility at the Bn-Bs interface. The TIP3P water model they employ is known to be too motile, and in such highly charged environments, polarization effects will have to be taken into account. It may thus be that in reality, water molecules at the interface will be highly localized.

pp. 32-38: While the authors present R² coefficients for matching experimental and tauRAMD predicted times, there are correlation coefficients such as the Spearman correlation that evaluates how well the ordering of unbinding times can be reproduced in the simulations. Please check if the Spearman correlation coefficients are maybe better than the presented R² values.

Fig. S1: Why did the authors perform a KS test against a Poisson CDF? This implies that unbinding follows a Poisson process. This may be the case for simple systems such as protein-ligand systems, but might break down in the case of more complicated process-underlying free energy landscapes and paths therethrough.

Reviewer #3 (Remarks to the Author):

This paper by the Wade group represents a very nice and interesting methodology for estimating residence (or unbinding times) for protein protein complexes, extending the reach of previously developed tau-RAMD.

The paper is well written, very clear. The methods are competently and thoroughly explained and the working hypothesis and results are consistent. Interestingly, the authors apply their approach to different protein systems, and observe a good quantitative correlation between their calculated times and experimental ones after residue mutations.

The whole approach is definitely worth publishing. To make their method/discussion of its reach more general, the authors may want to discuss the possibility of combining their approach with methods for the prediction of interaction surfaces for the generation of viable models of complexes, should the structure of the complex be unknown.

Responses to reviewers.

The comments of the reviewers are given followed by our responses in italics.

Reviewer #1 (Remarks to the Author):

This manuscript applied τ RAMD to investigate protein-protein unbinding pathways and estimate dissociation rates of the complexes. The work also compared τ RAMD, GaMD and metadynamics, and showed that τ RAMD is more computation efficiency. The authors performed pretty thorough tests with three protein-protein complexes: ribonuclease barnase (Bn) and barstar (Bs) wild-type and 22 single and double-point, 6 single-point mutants of bovine β -trypsin (BT) and its bovine pancreatic trypsin inhibitor (BPTI), and wild-type and 6 single-point mutants of bovine-chymotrypsin (BCT) and BPTI. The authors also collected 3 different exp from Bn-Bs and normalized them for easier comparison with computation results. The computation method is sound, and results are informative.

The tests for these extensive mutant datasets are useful for understanding the mutation effects and also demonstrated that τ RAMD's performance is impressive. The agreement between 28 computed and experimental data are very good. The method is robust and efficient. The reviewer found that the order of the manuscript is a bit hard to follow. It would be nice to see that τ RAMD yielded the same trend as experimentally measured koff earlier. The results nicely validate the method, but the reviewer had to find the figure near the end of the manuscript. Since the authors discussed the dissociation mechanisms, it would be important to show first that the computation agrees with experimental results. And then the discussion of the mechanisms would be more meaningful. It is fine to discuss the residence time later. but it may be helpful to move Figures 5 and 6 (either reporting koff or 1/koff) to near the beginning of Results, so it can be seen easily.

We thank the reviewer for their positive comments. We also thank the reviewer for the helpful suggestion to improve the readability of the manuscript which we have followed. In the revision, we moved “Residence time estimation for the protein-protein complexes” to section 3.1, “Determination of dissociation pathways for the Barnase-Barstar wild-type and mutant complexes” to section 3.2 and “Determination of the Unbinding Pathways for the wild-type and mutant BT-BPTI and BCT-BPTI Complexes” sections 3.3. In the marked-up text, we have highlighted only the titles of the three sections (rather than the complete sections) to indicate the changes in order. We have adjusted the numbering of the figures in the main text and the SI accordingly.

It is interesting to read that there are anchoring groups, e.g. clusters 1-3 in Figure 2A. However, what are these clusters exactly? Figures like 2A provides some information. But without more figures, it's hard to learn how they dissociate exactly and why these are important. Some representative figures for the pathways and/or the key clusters can be very useful.

We thank the reviewer for the suggestion. We have added a description in the Methods section under “Clustering” at line 268, as follows: “Clusters were then plotted and positioned on an increasing logarithmic scale of the average protein COM shift (Δ COM) in the cluster from the starting frame, reflecting dissociation from the bound (small Δ COM) to the unbound (large Δ COM) state. Each cluster contains frames with similar PP_REs content, and the size of the cluster indicates its population. Considering two clusters (C_i and C_j), the transition from one to the other,

i.e. ($C_i \rightarrow C_j$) - ($C_j \leftarrow C_i$), is indicated by arrows which collectively indicate the routes to unbinding.”

In addition, we have revised Figure 2 (now Figure 4) to facilitate comprehension of the unbinding pathways by showing a more detailed representation of the structures of the representative clusters. To do this, we show structures representative of the different clusters along the predominant unbinding pathway in WT and the alternative pathway in D35Abs. For a more complete view of the two unbinding mechanisms, we recommend viewing the two movies showing the representative main and alternative dissociation pathways for the WT and D35Abs complexes, respectively, which are available on Zenodo

(<https://doi.org/10.5281/zenodo.10728326>), as indicated in the caption of the figure. To further facilitate viewing these movies, we have uploaded the two movies as part of the Supplementary Information of the manuscript.

In addition, we have also modified Figures 4-6 by adding a legend below the IFP heatmaps to make it easier to identify the sites in the corresponding protein-protein interface and to follow the dissociation pathway.

Minor points.

line 75, what is WEMD?

Weighted ensemble MD (WEMD) was first introduced in line 61 using only the WE abbreviation with “MD”. We have now inserted the full name abbreviation at line 63.

line 98 abbreviation IFP appeared before the full name.

The abbreviation has now been moved after the full name (on line 103).

recommendation: publish after some revision.

Reviewer #2 (Remarks to the Author):

In their article „Computational screening of the effects of mutations on protein-protein off-rates and dissociation mechanisms by τ RAMD”, D’Arrigo et al. present the application of tauRAMD for the calculation of relative protein-protein unbinding rates. The authors test the approach on three different protein complexes and various mutants thereof.

While tauRAMD has initially been designed for the ranking of protein-ligand unbinding rates, the application for much more involving, and definitively represents a method development that deserves publication. As the authors state, such a tool is of interest for e.g. designing protein therapeutics with defined binding and unbinding kinetics. Performing a benchmarking with three different complexes and their mutants is commendable.

In general, I think the research presented is sound, and the manuscript is definitively worth publishing in CommsBio.

We thank the Reviewer for their positive comments.

However, I find some parts hard to read, especially the description of unbinding pathways, which appears to me quite lengthy and highly detailed, and will most likely

only be of interest for experts working exactly on the three presented protein/protein complexes. Given that tauRAMD is a method to firstly calculate unbinding times, I would like to ask the authors to restructure the manuscript and to first present the results on unbinding times. After having obtained the information of how well tauRAMD can predict such times, the interested reader then can delve into the intricacies of the unbinding paths (which might be needed to be shortened) and the role of water molecules therein.

We thank the Reviewer for this comment, which echoes that of Reviewer #1. We have reordered the manuscript as suggested, moving the section “Residence time estimation for the protein-protein complexes” to section 3.1, “Determination of dissociation pathways for the Barnase-Barstar wild-type and mutant complexes” to section 3.2 and “Determination of the Unbinding Pathways for the wild-type and mutant BT-BPTI and BCT-BPTI Complexes” to section 3.3. In the text, we have highlighted only the titles of the three sections, rather than the complete sections, to indicate the changes.

Specific remarks

p. 5, line 97: the formulation “no bias in the directions of the unbinding pathways” is misleading, as tauRAMD applies a bias force. I assume the authors want to state that no assumptions on the unbinding path need to be made beforehand.

Yes, that was the intention, we have rephrased the sentence accordingly (now on l100): “The random choice of the force orientation means that no a priori assumption on the direction of ligand unbinding is made, allowing sampling of diverse unbinding pathways”.

p. 8: Please point out that the pH values for protonation states were being chosen at 7.5 and 8 for reproducing conditions in the reference experiments.

We have now pointed out in lines 157 and 165 that the pH values were chosen to correspond to the experimental ones.

p. 10, line 191: how was the random force applied to the proteins? As a force on the center of mass that was then distributed to all atoms via mass weighing?

Yes. We have added the following sentence for clarification: “The random force was applied to the COM of the protein and distributed to all its atoms via mass-weighting.”

p. 12, line 249: please explain shortly what the Jaccard index is.

We have added a more complete description of the Jaccard Index used and revised the text as follows, at line 258:

“The derived contact matrix was then converted into a distance matrix using the Jaccard distance d_J , a coefficient that quantifies the dissimilarity between two sets. For each pair of RAMD frames, the associated d_J was given by the ratio of the number of common PP_REs contacts to the total number, subtracted from 1. As a

result, a d_J of 0 was associated with RAMD frames having the maximum identity in PP_RE content and vice versa for a d_J of 1. The derived distance matrix was finally clustered using the k-means algorithm implemented in the scikit-learn package⁴⁶ with default values for parameters.”

p. 22, 2nd paragraph: I think the authors need to be careful with their results on water motility at the Bn-Bs interface. The TIP3P water model they employ is known to be too motile, and in such highly charged environments, polarization effects will have to be taken into account. It may thus be that in reality, water molecules at the interface will be highly localized.

We understand the concern raised and acknowledge that the water model used is not polarizable and is therefore not fully representative of water behavior in such a highly electrostatic interprotein environment, as mentioned in lines 570-572. To ensure the reader to takes these results with caution, we have added the following note at the end of the paragraph on line 595: "However, it should be noted that the water model used for the simulations is not polarizable and may not fully account for the behavior of water in a highly polarizable environment such as the Bn-Bs interface.”

pp. 32-38: While the authors present R2 coefficients for matching experimental and tauRAMD predicted times, there are correlation coefficients such as the Spearman correlation that evaluates how well the ordering of unbinding times can be reproduced in the simulations. Please check if the Spearman correlation coefficients are maybe better than the presented R2 values.

Following the Reviewer's suggestion, we computed the Spearman correlation coefficient (ρ) for the computed to the experimental $1/k_{off}$ values for Bn-Bs (random force magnitude: 19kcal/mol/Å) and CBT/BT-BPTI (random force magnitude: 17kcal/mol/Å) for the different methods investigated. The ρ values are now given with the plots in Figures S3 and S8, respectively, showing mostly higher values than R2, and therefore supporting the existence of the correlations. However, as the intended use of the τ RAMD method is not only to rank the residence times but also to obtain the variation and show how similar or distant the values are, we believe that R2 values are more informative for our purposes.

Fig. S1: Why did the authors perform a KS test against a Poisson CDF? This implies that unbinding follows a Poisson process. This may be the case for simple systems such as protein-ligand systems, but might break down in the case of more complicated process-underlying free energy landscapes and paths therethrough.

We agree that sometimes unbinding may not follow a Poisson process but in many cases it does, and the KS test provides a helpful way to assess this. As the simulated dissociation events are independent, we expect that they should usually approach a Poisson statistic with sufficient sampling. Due to the greater complexity of protein-protein systems compared to the protein-small molecule systems, we ran a higher number of equilibration and RAMD trajectories to ensure a sufficiently large sampling. The two-sample KS test that we used quantifies the distance, D , between the empirical cumulative distribution function (ECDF), obtained from the dissociation

probability distribution observed in τ RAMD simulations, and the theoretical Poisson CDF with a specific time τ_{repl} . The small D values (<0.3) reported in Figure S1 for the KS test are indicative of the high similarity between the measured distribution of dissociation times and a Poisson distribution with a characteristic time τ_{repl} .

Reviewer #3 (Remarks to the Author):

This paper by the Wade group represents a very nice and interesting methodology for estimating residence (or unbinding times) for protein protein complexes, extending the reach of previously developed tau-RAMD.

The paper is well written, very clear. The methods are competently and thoroughly explained and the working hypothesis and results are consistent. Interestingly, the authors apply their approach to different protein systems, and observe a good quantitative correlation between their calculated times and experimental ones after residue mutations.

The whole approach is definitely worth publishing. To make their method/discussion of its reach more general, the authors may want to discuss the possibility of combining their approach with methods for the prediction of interaction surfaces for the generation of viable models of complexes, should the structure of the complex be unknown

We thank the Reviewer for their positive comments.

Regarding the last point for discussion, we have not yet validated the application of our approach with de novo predicted structures of protein-protein complexes, only with the predictions of the structures of complexes containing point mutations as described in the manuscript. However, the τ RAMD method has been successfully applied to complexes with small molecules for which it was necessary to generate a structure of the complex by docking or by comparative modeling (see e.g. Kokh et al., 2018).

As regards whether application of the τ RAMD protocol could be used to assess the stability of predicted protein-protein interfaces, we expect that it could but have not yet tested this. We have extended the last sentence in the Conclusion and Outlook to cover this application as follows: "We anticipate applications in the assessment of the stability of predicted protein-protein interfaces, in protein design and mutant screening,"

REVIEWERS' COMMENTS:

Reviewer #1 (Remarks to the Author):

The authors revised the manuscript accordingly. It is better and easier to read.

Reviewer #2 (Remarks to the Author):

My comments have been sufficiently addressed. I recommend publication.

Reviewer #3 (Remarks to the Author):

The authors have responded well to all the requests.

The paper is suitable for publication.